# Antimicrobial Activity of Apidermin 2 from the Honeybee *Apis mellifera*

**DOI:** 10.3390/insects13100958

**Published:** 2022-10-20

**Authors:** Bo-Yeon Kim, Yun-Hui Kim, Yong-Soo Choi, Man-Young Lee, Kwang-Sik Lee, Byung-Rae Jin

**Affiliations:** 1College of Natural Resources and Life Science, Dong-A University, Busan 49315, Korea; 2Department of Agricultural Biology, National Academy of Agricultural Science, Wanju 55365, Korea

**Keywords:** *Apis mellifera*, antimicrobial activity, apidermin, cuticular protein, honeybee

## Abstract

**Simple Summary:**

In the honeybee *Apis mellifera*, apidermin 2 (APD 2) is known as a cuticular protein. However, the antimicrobial properties of *A. mellifera* APD 2 (AmAPD 2) have not been characterized. Herein, we provide the first demonstration that AmAPD 2 exhibits antibacterial and antifungal activities. We found that AmAPD 2 induced structural damage by binding to bacterial and fungal cell walls, indicating that AmAPD 2 has the antimicrobial action of an antimicrobial peptide. Our findings demonstrate a novel role of AmAPD 2 as an antimicrobial agent in honeybees.

**Abstract:**

Apidermins (APDs) are known as structural cuticular proteins in insects, but their additional roles are poorly understood. In this study, we characterized the honeybee, *Apis mellifera*, APD 2 (AmAPD 2), which displays activity suggesting antimicrobial properties. In *A. mellifera* worker bees, the *AmAPD 2* gene is transcribed in the epidermis, hypopharyngeal glands, and fat body, and induced upon microbial ingestion. Particularly in the epidermis of *A. mellifera* worker bees, the *AmAPD 2* gene showed high expression and responded strongly to microbial challenge. Using a recombinant AmAPD 2 peptide, which was produced in baculovirus-infected insect cells, we showed that AmAPD 2 is heat-stable and binds to live bacteria and fungi as well as carbohydrates of microbial cell wall molecules. This binding action ultimately induced structural damage to microbial cell walls, which resulted in microbicidal activity. These findings demonstrate the antimicrobial role of AmAPD 2 in honeybees.

## 1. Introduction

The insect cuticle serves as an exoskeleton with multifunctional biological properties. It is mainly composed of structural proteins and chitins [1,2]. Cuticular proteins (CPs) consist of a variety of structural proteins, and the epidermis is the primary site for synthesizing the CPs [3]. Arthropod CP families have been categorized by whole-genome sequencing and described based on characteristics and phylogenetic analysis [2].

Among the CP families from honeybees [3,4,5,6], genes of the apidermin (APD) family are known in the honeybee *Apis mellifera* and are named *APD 1*, *APD 2*, and *APD 3* [4]. *A. mellifera* APDs are highly hydrophobic and APD 1 and APD 2 possess an arginine-rich motif: RERR. The genes encoding APD 2 and APD 3 are expressed in various tissues of *A.*
*mellifera*. Specifically, APD 2 is a component of internal flexible cuticles [4], and proteomic analyses have demonstrated that APD 2 is also detected in the fat body of *A. mellifera* worker bees [7] and is an exoskeletal component [8]. Cuticular proteins, including APD 2, showed an increased abundance in summer *A. mellifera* worker bees in comparison to winter worker bees [9]. In addition, *APD* genes have been analyzed through genome-wide identification in wasps [2,10,11] and bumblebees [12].

Although APDs are known as structural cuticular proteins, the additional roles of APDs remain largely unknown. A previous study revealed that *APD 2* in bumblebees was differentially expressed during parasite exposure, and its expression pattern was correlated with antimicrobial peptides (AMPs) known to exist in bees [13]. This thus suggests that APD 2 may be a novel AMP [13]. Moreover, our previous study comparing microbiome and RNA-sequencing analyses of honeybees (*Apis cerana*) susceptible and resistant to sacbrood virus disease revealed that the gut microbiome was different between the two strains [14], and *APD 2* was more highly expressed in the resistant than the susceptible strains. Therefore, we aimed to elucidate the novel role of APD 2 in honeybees and demonstrate its antimicrobial activity.

Herein, we provide the first demonstration that *A. mellifera* APD 2 (AmAPD 2) exhibits antimicrobial activity. Because APD 2 is highly hydrophobic [4,13], which is a common feature of AMPs [15], we hypothesized that AmAPD 2 serves as an AMP in honeybees. In addition, we investigate the antimicrobial mechanism of AmAPD 2, and notably, we demonstrate that AmAPD 2 exhibits antimicrobial action like that of an AMP.

## 2. Materials and Methods

### 2.1. Honeybees

*Apis mellifera* honeybees were obtained from an apiary at Dong-A University, Busan, Republic of Korea. Newly-emerged, 1-day-old worker bees were marked on the thorax using a paint marker [16]. The marked *A. mellifera* worker bees were then placed back into the hive and maintained until experimentation.

### 2.2. Peptide Sequence Analysis

To compare AmAPD 2 to the APD 2 proteins previously found in other bee species, the deduced peptide sequences from the *APD 2* genes were aligned using MacVector (ver. 6.5, Oxford Molecular Ltd., Oxford, UK). *Apis mellifera* APD 2 (GenBank accession no. NM_001085346), *A. cerana* APD 2 (GenBank accession no. XM_017050089), and *Bombus terrestris* APD 2 (GenBank accession no. XM_003394905) were used in this study.

### 2.3. RNA Extraction, cDNA Synthesis, and Quantitative Reverse Transcription-PCR (qRT-PCR)

Total RNA was isolated from the epidermis, fat body, and hypopharyngeal glands of *A. mellifera* worker bees using RNAiso Plus (TaKaRa Bio, Shiga, Japan). The concentration and purity of each RNA sample were analyzed using a NanoDrop One (Thermo Fisher Scientific, Madison, WI, USA). Next, cDNA was synthesized using 2 μg of RNA per sample with the AccuPower RT PreMix (BIONEER, Daejeon, Korea). Relative expression of the *AmAPD 2* gene was measured using qRT-PCR on a CronoSTAR^TM^ 96 real-time PCR system (Clontech, Palo Alto, CA, USA) with TB Green Premix Ex Taq^TM^ (TaKaRa Bio) in 25 μL reactions, each containing 80 ng of cDNA and 0.2 nM of primers: forward, 5′–GTTAATCCTCTTCGCCATCGT–3′ and reverse, 5′–GGCAATAGTGGGTGCAAGA–3′. The PCR protocol consisted of an initial denaturation step at 95 °C for 5 min followed by 40 cycles of 95 °C for 15 s and 60 °C for 30 s. The gene expression level of *AmAPD 2* was normalized to that of the internal control gene *actin* (XM_003251416), which was amplified using the forward primer, 5′–ATGTGTGACGACGAAGTAGCA–3′, and the reverse primer, 5′–TCCTTTTGACCCATACCG–3′. The qRT-PCR experiment was performed with three biological replicates and analyzed using the 2^−ΔΔCT^ method [17].

### 2.4. Microbial Feeding Experiment

Six-day-old *A. mellifera* worker bees were treated with heat-killed pathogens as described in our previous study [16]. The bee pathogens used in this study were *Paenibacillus larvae* and *Ascosphaera apis* [16,18], each used as separate treatments. The worker bees were maintained in cages (11.3 × 7.0 × 4.3 cm) in an incubator at 34 °C with 80% humidity and fed a 40% sucrose solution with or without heat-killed *P. larvae* (2.5 × 10^2^ cells per bee) or *A. apis* (5 × 10^3^ cells per bee) over a 24 h period. For each treatment, 40 bees (*n* = 40) were used, and the experiment was performed in triplicate.

### 2.5. Production and Purification of Recombinant AmAPD 2

Recombinant AmAPD 2 was produced in *Spodoptera frugiperda* (Sf9) insect cells using a baculovirus expression vector system [19]. First, total RNA was extracted from the whole body of *A. mellifera* worker bees using TRIzol reagent (Invitrogen, Carlsbad, CA, USA). From this, cDNA was synthesized, which was then PCR-amplified. The primers used for amplification included restriction enzyme sites and a His-tag sequence in the reverse primer: the forward primer, 5′–AGATCTATGAAATCCCTGTTAATCC–3′, included a *Bgl* II restriction enzyme site (double-underlined) and the reverse primer, 5′–TCTAGATTAATGATGATGATGATGATGCCATGCTTTCCAC–3′, included an *Xba* I restriction enzyme site (double-underlined) and a His-tag sequence (underlined). This PCR product was inserted into the *pBacPAK8* vector (Clontech, Palo Alto, CA, USA). The expression vector construct (*pBacPAK8-AmAPD 2*) was co-transfected with the baculoviral DNA [19] into Sf9 insect cells to produce recombinant baculoviruses expressing recombinant AmAPD 2. The cultured medium was incubated for 5 days then harvested and centrifuged at 10,000× *g* for 10 min to remove cell debris. The supernatant was precipitated in 1 M ammonium sulfate and centrifuged at 15,000× *g* for 20 min. The pelleted proteins were resuspended in phosphate-buffered saline (PBS: 140 mM NaCl, 27 mM KCl, 8 mM Na_2_HPO_4_, 1.5 mM KH_2_PO_4_, pH 7.4). The recombinant AmAPD 2 was purified using the MagneHis^TM^ Protein Purification System (Promega, Madison, WI, USA) according to the manufacturer’s instructions and then quantified using a Bio-Rad protein assay kit (Bio-Rad, Hercules, CA, USA).

### 2.6. Sodium Dodecyl Sulfate-Polyacrylamide gel Electrophoresis (SDS-PAGE) and Western Blot Analysis

Protein samples were analyzed by SDS-PAGE on a 12% gel followed by Western blot analysis using an enhanced chemiluminescence Western blot system (Amersham Biosciences, Piscataway, NJ, USA). An anti-His-tag antibody (1:10,000 [*v*/*v*]) (Abcam, Cambridge, UK) was used as a primary antibody, and a horseradish-peroxidase-conjugated anti-rabbit IgG antibody (1:5000 [*v*/*v*]) was used as a secondary antibody. Exposure and detection procedures were performed according to the manufacturer’s instructions.

### 2.7. Binding of AmAPD 2 to Carbohydrates

To measure the ability of AmAPD 2 to bind to carbohydrates, a carbohydrate-binding assay for the recombinant AmAPD 2 was performed using lipopolysaccharide (LPS), mannan, and *N*-acetyl-D-glucosamine as described in our previous study [20]. The carbohydrates were purchased from Sigma-Aldrich (St. Louis, MO, USA). Ninety-six-well plates coated with 0.2 mM/well of LPS, mannan, or *N*-acetyl-D-glucosamine were incubated with 0.1 mg/mL bovine serum albumin in 50 mM Tris-HCl buffer (pH 8.0) for 2 h and then incubated at 25 °C for 3 h with 100 μL recombinant AmAPD 2 at concentrations of 0, 100, 200, 300, 400, or 500 nM per well. After washing, the plates were incubated with the rabbit anti-His-tag antibody (1:10,000 [*v*/*v*]) and then with a horseradish-peroxidase-conjugated goat anti-rabbit IgG antibody (1:5000 [*v*/*v*], Enzo Life Sciences, Farmingdale, NY, USA), each at 25 °C for 1 h. Finally, the plates were incubated with 100 μL/well of 3,3′,5,5′-tetramethyl-benzidine substrate solution at 25 °C for 10 min, and the reaction was then stopped using 100 μL/well of 2 M H_2_SO_4_. To quantify carbohydrate-bound protein, absorbance was measured at 450 nm using a microplate reader (Bio-Rad Model 3550).

The thermal stability of recombinant AmAPD 2 was determined at varying temperatures, ranging from 40 °C to 70 °C, using a peptidoglycan (PG)-binding assay. Recombinant AmAPD 2 was preincubated at 40 °C, 50 °C, 60 °C, and 70 °C for 1 h, and added to each well (100 nM per well) of a 96-well plate coated with 0.2 mM/well of PG from *Bacillus subtilis* (Sigma-Aldrich). The binding assay was performed as described above.

### 2.8. Microbial Binding Assay

A microbial binding assay for the recombinant AmAPD 2 against live bacteria and fungi was performed with microorganisms used in our previous studies [20,21,22,23]. The gram-negative bacterium *Escherichia coli* DH5α, gram-positive bacterium *Bacillus thuringiensis* 656-3, and entomopathogenic fungus *Beauveria bassiana* SFB-205 were incubated with the recombinant AmAPD 2 (0.8 μg) at 25 °C for 10 min. The microbial samples were centrifuged at 2500× *g* for 5 min. The supernatants (free AmAPD 2) and pellets (bound AmAPD 2) were subjected to SDS-PAGE on a 12% gel followed by Western blot analysis using an anti-His-tag antibody (1:10,000 [*v*/*v*], Abcam, Waltham, MA, USA) as described above.

A similar microbial binding assay using recombinant AmAPD 2 heat treated at 50 °C for 1 h was performed using live *B. thuringiensis*. The supernatant and pellet samples were obtained as described above. In addition, pellets were washed with 30 μL of 50 mM NaCl in PBS and 150 mM NaCl in PBS, sequentially, prior to SDS-PAGE and Western blot procedures, which were performed as described above but with the supernatant from the second washing step (150 mM NaCl in PBS) used as a control [24].

### 2.9. Immunofluorescent Staining

For immunofluorescent staining, microbial samples of *E. coli* DH5α, *B. thuringiensis* 656-3, and *B. bassiana* SFB-205 were treated with recombinant AmAPD 2, as described in the binding assay, and then incubated with the rabbit anti-His-tag antibody (1:500 [*v*/*v*]) at 25 °C for 1 h. After washing, the microbial samples were probed with a secondary fluorescein-conjugated goat anti-rabbit antibody (1:400 [*v*/*v*]; Santa Cruz Biotech, Inc., Santa Cruz, CA, USA) at 25 °C for 1 h. Images were obtained using a confocal microscope (Carl Zeiss LSM 510, Zeiss, Jena, Germany).

### 2.10. Scanning Electron Microscopy (SEM)

For SEM, microbial samples were incubated with or without 0.8 μg/well of recombinant AmAPD 2 in 96-well plates. *Escherichia coli* and *B. thuringiensis* cells were incubated at 37 °C and *B. bassiana* conidia were incubated at 22 °C, all for 24 h with shaking at 220 rpm. After fixation with 2.5% glutaraldehyde (Sigma-Aldrich) at 25 °C for 15 min, the samples were dehydrated and coated with gold. Images were obtained using SEM (Hitachi S-530 SEM, Hitachi, Japan).

### 2.11. Antimicrobial Activity Assay

The antimicrobial activity assay for the recombinant AmAPD 2 was evaluated according to the liquid growth inhibition method described in our previous studies [20,21,22,23]. Microbial samples were incubated with 0.8 μg/well of recombinant AmAPD 2 in 96-well plates. Incubation conditions differed between organisms: *E. coli* (2 × 10^3^ cells per well) and *B. thuringiensis* (2 × 10^3^ cells per well) cells were incubated at 37 °C for 24 h and *B. bassiana* conidia (1 × 10^3^ conidia per well) were incubated at 22 °C for 48 h, all shaken at 220 rpm. The growth inhibition of microorganisms was measured using a microplate reader with absorbance at 595 nm. The antimicrobial activity of the recombinant AmAPD 2 was assessed using the minimal inhibitory concentration (MIC_50_) of 50% growth inhibition (*E. coli* and *B. thuringiensis*) and the half-maximal concentration (IC_50_) of 50% growth inhibition (*B. bassiana*). Experiments were performed with three independent replicates, and data are presented as the mean ± standard deviation (SD).

### 2.12. Statistical Analysis

The data was analyzed using independent unpaired 2-tailed Student’s *t*-tests comparing treatments to controls (Statistical software SPSS PASW 22.0 package for Windows, IBM, Chicago, IL, USA). An alpha level of α = 0.05 was used to test statistical significance, which is presented using asterisks: ** *p* < 0.01 and * *p* < 0.05.

## 3. Results

### 3.1. Expression Profile of AmAPD 2 in A. mellifera Worker Bees

While characterizing transcriptome profiles of honeybees (*A. cerana*) susceptible and resistant against sacbrood virus disease [14], we obtained an APD 2 gene with a relatively high-fold increase in expression in resistant *A. cerana* worker bees. Based on the differential expression of this *A. cerana APD 2* gene, we cloned a cDNA encoding *A. mellifera* APD 2 (AmAPD 2) for further investigation. Peptide sequence analysis revealed that AmAPD 2 exhibits high similarity to the *A. cerana* APD 2 (90% peptide sequence identity) and the bumblebee *B. terrestris* APD 2 (65% peptide sequence identity) (Figure 1A). Considering this high similarity, and the known properties of APD 2, we focused on the antimicrobial role of AmAPD 2 in this study.

We examined the expression profile of *AmAPD 2* in *A. mellifera* worker bees using qRT-PCR (Figure 1B). In addition to the tissues known to exhibit *APD 2* expression in honeybees, we selected tissues involved in antimicrobial actions because we hypothesized that AmAPD 2 serves a role as an AMP in honeybees. We found that *AmAPD 2* was expressed in the epidermis, hypopharyngeal glands, and fat body of *A. mellifera* worker bees. The expression level of *AmAPD 2* was particularly high in the epidermis.

Because APD 2 possesses AMP-like properties [13], we assessed whether *AmAPD 2* expression is induced in the tissues of *A. mellifera* worker bees upon microbial challenge. We examined the *AmAPD 2* transcription patterns in the epidermis, hypopharyngeal glands, and fat body of bees following microbial ingestion of the heat-killed bee pathogens *P. larvae* and *A. apis*. Notably, gene expression analysis showed that *AmAPD 2* expression was significantly increased in all the tested tissues, most distinctively the epidermis (Figure 2).

### 3.2. Antimicrobial Activity of Recombinant AmAPD 2

To investigate the antimicrobial activity of AmAPD 2, we produced a recombinant AmAPD 2 peptide (Figure 3A) in baculovirus-infected insect cells, which secrete the mature peptide after signal sequence cleavage [25]. Using the recombinant AmAPD 2, we determined whether AmAPD 2 binds to the carbohydrates LPS, mannan, and *N*-acetyl-D-glucosamine, all of which are known from microbial cell walls. The carbohydrate-binding assay revealed that AmAPD 2 recognizes and binds to these carbohydrates (Figure 3B).

In addition, the thermal stability of recombinant AmAPD 2 was determined by binding heat-treated AmAPD 2 to PG. The result of the PG-binding assay revealed that recombinant AmAPD 2 was stable at heats ranging from 40 °C to 70 °C for 1 h (Figure 3C).

We performed a microbial binding assay using the recombinant AmAPD 2 against the live bacteria and fungi used in several of our previous studies [20,21,22,23]. The microbial binding assay using western blot analysis revealed that the recombinant AmAPD 2 binds to *E. coli*, *B. thuringiensis*, and *B. bassiana* (Figure 4A). We also found that recombinant AmAPD 2 heat treated at 50 °C for 1 h still bound to *B. thuringiensis* (Figure 4A). Moreover, we observed the localization of the recombinant AmAPD 2 on the microbial surfaces using immunofluorescent staining (Figure 4B).

To determine whether the microbial cell wall binding of the recombinant AmAPD 2 leads to antimicrobial activity due to structural damage, we treated live microorganisms with the recombinant AmAPD 2 and observed their cell walls using SEM. The result showed that the recombinant AmAPD 2 induced structural damage (Figure 5), which resulted in antimicrobial activity (Table 1).

## 4. Discussion

In one of our previous studies, we found differences between the gut microbiomes of *A. cerana* strains susceptible to and resistant against sacbrood virus disease [14] and that *APD 2* expression increased in resistant *A. cerana* worker bees. Furthermore, the differential expression of *APD 2* in parasite-exposed *B. terrestris* was strongly correlated with the expression of AMPs [13]. We also know that APD 2 is highly hydrophobic [4,13], which, again, is a trait seen with AMPs [15]. These results suggest that APD 2 may function as an antimicrobial agent in bees. Here, we found that AmAPD 2 indeed exhibited microbicidal activities against both bacteria and fungi.

In bees, APD 2 is a cuticular protein that is expressed primarily in the trachea and gut [4]. Proteomic analysis revealed that APD 2 is detected also in the cuticle, epidermis, and fat body of adult *A. mellifera* workers [7]. In this study, we found that *AmAPD 2* is expressed in the epidermis and fat body of *A. mellifera* worker bees, but also in the hypopharyngeal glands, and with a particularly high expression levels in the epidermis. Considering that the epidermis is the primary expression tissue of CPs [3,8] and that AMPs are expressed in the fat body and hypopharyngeal glands of *A. mellifera* worker bees [16], our results suggest that AmAPD 2 may function as both an antimicrobial agent as well as a CP in bees.

From these results, we hypothesized that *AmAPD 2* is involved in responses to microbial challenges in *A. mellifera* worker bees. In the present study, *AmAPD 2* gene expression increased in the epidermis, hypopharyngeal glands, and fat body following microbial ingestion. Notably, this suggests that AmAPD 2 is involved in the innate immune response of *A. mellifera* worker bees.

Because of this increased expression, we also hypothesized that AmAPD 2 acts as an antimicrobial agent. In insects, AMPs and AMP-like peptides with antimicrobial functions induce structural damage by binding to microbial cell walls [20,21,22,23,26,27,28]. Therefore, we tested whether AmAPD 2 would exhibit similar antimicrobial action. We found that AmAPD 2 exhibits antibacterial and antifungal activities, and that this antimicrobial activity was indeed due to the structural damage caused by microbial binding. Moreover, our PG-binding assay of heat-treated recombinant AmAPD 2 revealed that it was stable at high temperatures, another trait seen with AMPs [29,30,31,32]. In bees, AMP-like peptides—such as serine protease inhibitors [20,22], secapins [21], and major royal jelly proteins [18,22,28]—show an additional antimicrobial role as AMPs. These peptides apparently behave as broad-spectrum antimicrobial agents in bees. In this study, we assayed the antimicrobial activity of AmAPD 2 using the recombinant AmAPD 2 with 6× His-tag residues at the C-terminus. Our previous studies revealed that His-tag residues in the recombinant peptide do not affect antimicrobial activity [20,21,22,23,33]. Thus, our data demonstrates that AmAPD 2 can function as an antimicrobial peptide.

In addition, *APD 2* expression in *B. terrestris* is potentially a direct defense or a response to repair the gut damage caused by parasites [13]. An interesting aspect of AmAPD 2 is that it may also function as a barrier to invaders, such as parasites and pathogens. Collectively, our findings demonstrate the novel role of AmAPD 2 as an antimicrobial agent and a cuticular protein in honeybees.

## 5. Conclusions

Our findings provide the first evidence that AmAPD 2 exhibits antimicrobial activity, demonstrating that AmAPD 2 functions as an antimicrobial agent as well as a cuticular protein in *A. mellifera* worker bees. In addition, AmAPD 2 has the same antimicrobial action as AMPs and AMP-like peptides. These results provide novel insights into APDs to better understand the functional role of CPs in bees.

## Figures and Tables

**Figure 1 insects-13-00958-f001:**
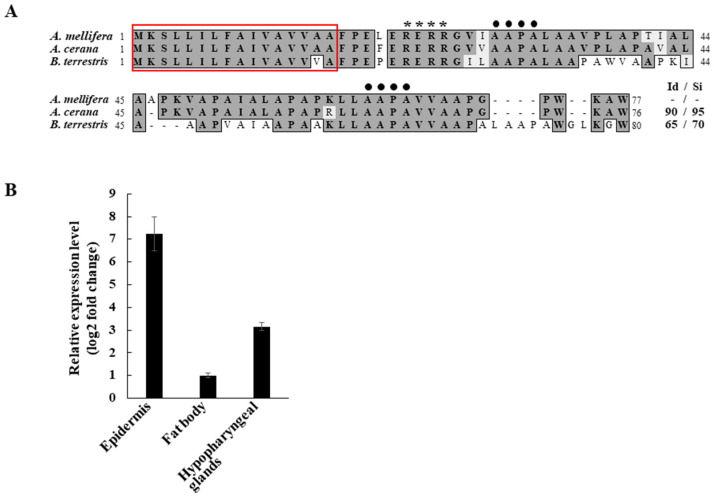
The amino acid sequence and tissue distribution of *Apis mellifera* apidermin 2 (AmAPD 2). (**A**) Alignment of the amino acid sequences of bee apidermin 2 (APD 2). The predicted signal sequences are boxed [4,13]. The conserved arginine-rich motifs and hydrophobic tetra peptides are shown by asterisks and solid circles, respectively [4]. The GenBank accession numbers of the aligned sequences are AmAPD 2 (NM_001085346), *A. cerana* APD 2 (XM_017050089), and *B. terrestris* APD 2 (XM_003394905). The identity/similarity (Id/Si) values were obtained using the AmAPD 2 sequence as a reference. (**B**) Expression of *AmAPD 2* in the epidermis, fat body, and hypopharyngeal glands of *A. mellifera* worker bees, as assessed via quantitative reverse transcription-PCR (qRT-PCR) (*n* = 20).

**Figure 2 insects-13-00958-f002:**
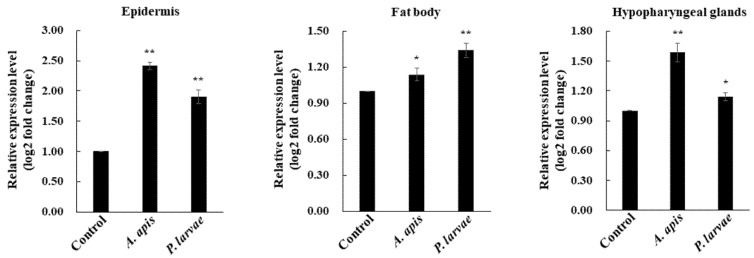
Transcription pattern of *Apis mellifera apidermin 2* (*AmAPD 2*) in *A. mellifera* worker bees upon microbial ingestion. The transcription pattern of *AmAPD 2* was analyzed in *A. mellifera* worker bees fed *A. apis* or *P. larvae* over 24 h. Untreated *A. mellifera* worker bees were used as controls. Total RNA was extracted from the epidermis, fat body, and hypopharyngeal glands of *A. mellifera* worker bees (*n* = 40). *AmAPD 2* transcription was analyzed using quantitative reverse transcription-PCR (qRT-PCR). Data are represented as the mean ± standard deviation (SD) of three replicates (* *p* < 0.05 and ** *p* < 0.01).

**Figure 3 insects-13-00958-f003:**
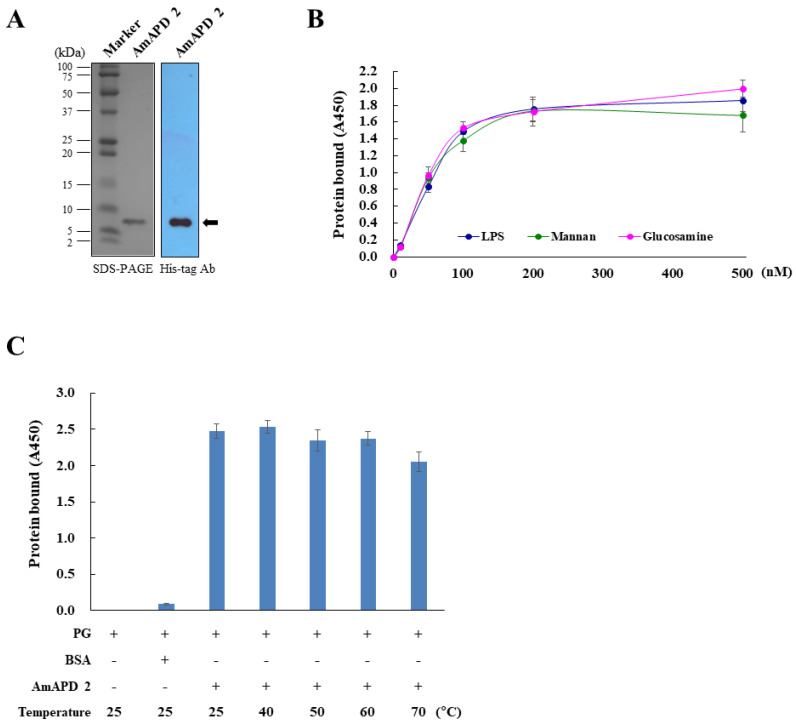
Carbohydrate-binding of recombinant *Apis mellifera* apidermin 2 (AmAPD 2). (**A**) Production of recombinant AmAPD 2 in baculovirus-infected Sf9 insect cells. The purified recombinant AmAPD 2 was verified via 12% sodium dodecyl sulfate-polyacrylamide gel electrophoresis (SDS-PAGE, left), followed by western blot analysis using an anti-His-tag antibody (right). The molecular weight standard and recombinant AmAPD 2 are indicated. (**B**) Binding of recombinant AmAPD 2 to lipopolysaccharide (LPS), mannan, and *N*-acetyl-D-glucosamine (*n* = 3). (**C**) Binding of heat-treated recombinant AmAPD 2 to peptidoglycan (PG) (*n* = 3). Recombinant AmAPD 2 was preincubated at 40 °C–70 °C for 1 h.

**Figure 4 insects-13-00958-f004:**
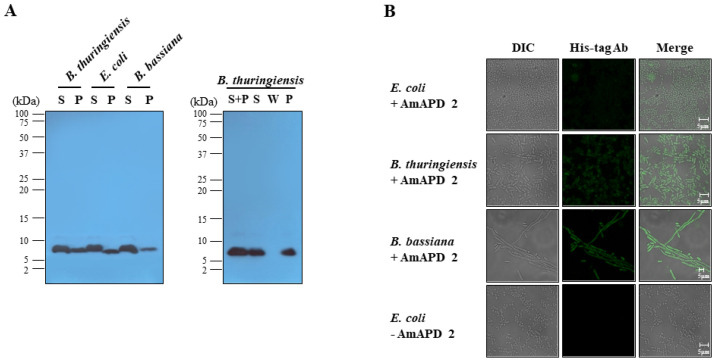
The binding ability of recombinant *Apis mellifera* apidermin 2 (AmAPD 2) to microbial cell walls. (**A**) Western blot analysis of the binding of recombinant AmAPD 2 to microbial cell walls. Live *E. coli*, *B. thuringiensis*, or *B. bassiana* cells were treated with the recombinant AmAPD 2 for 10 min. The pellet (P) and supernatant (S) represent AmAPD 2 bound to the microbial pellet and free AmAPD 2 in the supernatant, respectively (Left panel). Live *B. thuringiensis* cells were treated with the recombinant AmAPD 2 that had been preincubated at 50 °C for 1 h (Right panel). As a control in the heat-pretreatment experiment, the wash sample (W) used was the supernatant in the second washing step of the pellet. Western blotting was performed using an anti-His-tag antibody. (**B**) Immunofluorescence staining of *E. coli*, *B. thuringiensis*, or *B. bassiana* cells showing the binding of recombinant AmAPD 2 (green) to the microorganisms’ cell walls. Staining was performed using an anti-His-tag antibody. For a negative control, *E. coli* cells pre-incubated without recombinant AmAPD 2 were used (*E. coli*-AmAPD 2). Merged confocal images are shown in which scale bars = 5 μm.

**Figure 5 insects-13-00958-f005:**
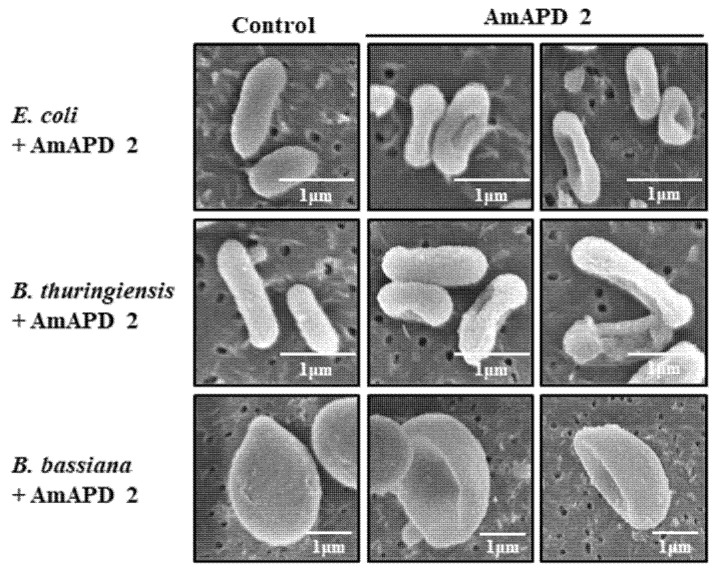
Scanning electron microscopy (SEM) observation of recombinant *Apis mellifera* apidermin 2 (AmAPD 2)-induced structural damage to microbial cell walls. The control column shows untreated *E. coli*, *B. thuringiensis*, or *B. bassiana* samples next to two AmAPD 2 columns showing treated microbial samples. Scale bars = 1 μm.

**Table 1 insects-13-00958-t001:** Antimicrobial activity of recombinant *Apis mellifera* apidermin 2 (AmAPD 2) against bacteria and fungi.

	Microorganism	MIC_50_ (µM)
Gram-positive bacterium	*B. thuringiensis*	9.59 ± 0.24
Gram-negative bacterium	*E. coli*	7.90 ± 0.12
		IC_50_ (µM)
Entomopathogenic fungus	*B. bassiana*	23.8 ± 0.35

## Data Availability

Not applicable.

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
