# Peer review of "Antimicrobial Activity of Apidermin 2 from the Honeybee Apis mellifera"

_insects, 2022, doi:10.3390/insects13100958_

Round 1
Reviewer 1 Report
Dear authors,
I have gone through the whole manuscript thoroughly; your work is commendable covering relevant literature. It is an excellent piece of work, from this MS, the authors characterized the AmAPD 2 peptide from the honeybee and find out the antimicrobial potentiality of the peptide. However, the English language in the manuscript needs to recheck thoroughly so that the information and contents of the ms can be expressed adequately. I suggest the authors to re-write the abstract part and reconstruct a few sentences in the introduction part and the rest of the MS.
1. In the simple summary section, the authors need to reconstruct the sentence in lines 23-24, “We found that AmAPD 2 has the 23 antimicrobial action of an antimicrobial peptide and induces structural damage by 24 binding to microbial cell walls”.
2. Abstract and introduction parts sentence construction need to be improved for better understanding. E.g., Line no 53 “In honeybees, CP genes have been identified”.
Author Response
Reviewer 1
I have gone through the whole manuscript thoroughly; your work is commendable covering relevant literature. It is an excellent piece of work, from this MS, the authors characterized the AmAPD 2 peptide from the honeybee and find out the antimicrobial potentiality of the peptide. However, the English language in the manuscript needs to recheck thoroughly so that the information and contents of the ms can be expressed adequately. I suggest the authors to re-write the abstract part and reconstruct a few sentences in the introduction part and the rest of the MS.
- In the simple summary section, the authors need to reconstruct the sentence in lines 23-24, “We found that AmAPD 2 has the 23 antimicrobial action of an antimicrobial peptide and induces structural damage by 24 binding to microbial cell walls”.
Author’s response
We are very grateful for your valuable comments. According to reviewer #1’s comment, we revised it.
- Abstract and introduction parts sentence construction need to be improved for better understanding. E.g., Line no 53 “In honeybees, CP genes have been identified”.
Author’s response
We are very grateful for your valuable comments. According to reviewer #1’s comment, we revised it.
Reviewer 2 Report
Authors performed an interesting study on a cuticular protein APD2 in worker bees, they characterized the APD2 gene identified from transcriptome profiles of honeybee susceptible and resistant to sacbrood virus, found APD2 as a cuticular protein could bind to carbohydrates and bacteria and fungi, it damaged the pathogen cell wall and showed antimicrobial activity like AMPs. The study was well designed and performed, it may provide a new candidate for resistant honeybee selection and enrich the antimicrobial agents in insects. I only have some comments that might improve the manuscript quality, listed below:
1. In the binding assay and antimicrobial activity assay, what is the control used? BSA or something else? I have a concern about these assays, although the APD2 were purified using His beads, is it possible the purified APD2Sf9 contaminated with Sf9 produced antimicrobial peptides or any other antimicrobial agents? If you would like to use Sf9 medium or Sf9 produced His tagged small peptide as control to test these assays, the binding assay and antimicrobial assay will be more rigorous.
2. In the binding assay, you used LPS, mannan and N-acetyl-D-glucosamine to test the APD2 binding activity, while the cell wall components in different pathogens such as Gram+ and Gram- bacteria, vary a lot, and PGNs in bacterial cell wall are rich, whether APD2 can Bind to PGN like the binding assay in the experiment https://doi.org/10.1016/j.dci.2014.01.017?
3. Total APD2 amount (S+P) before binding assay should be in Figure 4A to compare APD2 in P and S with the total APD2.
4. The first 16 AAs is predicted to be a signal peptide, while based on your primers, the full length of APD2 was expressed in Sf9 cells, so that means the mature APD2 should be secreted into medium, in your method, I didn’t find where was your purified APD2 from, please detail the purification process.
5. Please indicate the restriction enzyme sites in the primers in lin127 and 128.
6. Please add “wash step” in section 2.7, I believe you did, but the information is missing in your text.
7. APD2 is highly hydrophobic as seen in AMPs, could you analyze their structures using some software and check if they share similar structure properties?
8. Usually, the AMPs are heat-stable, and APD2 has a lot of similar features of AMPs, so you may test APD2 heat stability, if it is stable and still has antimicrobial activity, APD2 may be a relative of AMPs.
9. It would be better to add the BSA control group in Figure 4B binding assay?
10. There are so many names in ref 10 and 12, can you just list the first 3 plus et al.?
11. Please add statistical significance information in figure2 legend.
12. Do all the APD2 in Figure 1 have signal peptides? I just check AmAPD2, please check another two, and add the signal peptide information in figure 1A.
13. Line 33, delete “its expression is”.
14. In drone bee and queen, can pathogens upregulate APD2 expression?
Author Response
Reviewer 2
Authors performed an interesting study on a cuticular protein APD2 in worker bees, they characterized the APD2 gene identified from transcriptome profiles of honeybee susceptible and resistant to sacbrood virus, found APD2 as a cuticular protein could bind to carbohydrates and bacteria and fungi, it damaged the pathogen cell wall and showed antimicrobial activity like AMPs. The study was well designed and performed, it may provide a new candidate for resistant honeybee selection and enrich the antimicrobial agents in insects. I only have some comments that might improve the manuscript quality, listed below:
In the binding assay and antimicrobial activity assay, what is the control used? BSA or something else? I have a concern about these assays, although the APD2 were purified using His beads, is it possible the purified APD2Sf9 contaminated with Sf9 produced antimicrobial peptides or any other antimicrobial agents? If you would like to use Sf9 medium or Sf9 produced His tagged small peptide as control to test these assays, the binding assay and antimicrobial assay will be more rigorous.
Author’s response
We are very grateful for your valuable comments. According to reviewer #2’s comment, we revised the sentences in the M&M section 2.5 and Discussion section.
- In the binding assay, you used LPS, mannan and N-acetyl-D-glucosamine to test the APD2 binding activity, while the cell wall components in different pathogens such as Gram+ and Gram- bacteria, vary a lot, and PGNs in bacterial cell wall are rich, whether APD2 can Bind to PGN like the binding assay in the experiment https://doi.org/10.1016/j.dci.2014.01.017?
Author’s response
We are very grateful for your valuable comments. According to reviewer #2’s comment, we performed the experiment with a PGN (M&M section 2.7 and Fig. 3C) and cited the reference suggested by reviewer.
- Total APD2 amount (S+P) before binding assay should be in Figure 4A to compare APD2 in P and S with the total APD2.
Author’s response
We are very grateful for your valuable comments. According to reviewer #2’s comment, we performed the binding experiment (M&M section 2.8 and Fig. 4A-Lower panel) and cited the reference suggested by reviewer.
- The first 16 AAs is predicted to be a signal peptide, while based on your primers, the full length of APD2 was expressed in Sf9 cells, so that means the mature APD2 should be secreted into medium, in your method, I didn’t find where was your purified APD2 from, please detail the purification process.
Author’s response
We are very grateful for your valuable comments. According to reviewer #2’s comment, we revised the sentences in the M&M section 2.5 and Result section.
- Please indicate the restriction enzyme sites in the primers in lin127 and 128.
Author’s response
We are very grateful for your valuable comments. According to reviewer #2’s comment, we added the restriction enzyme sites in the M&M section 2.5.
- Please add “wash step” in section 2.7, I believe you did, but the information is missing in your text.
Author’s response
We are very grateful for your valuable comments. According to reviewer #2’s comment, we performed the binding experiment (M&M section 2.8 and Fig. 4A-Lower panel).
- APD2 is highly hydrophobic as seen in AMPs, could you analyze their structures using some software and check if they share similar structure properties?
Author’s response
We are very grateful for your valuable comments. According to reviewer #2’s comment, we revised the sentences in the legend of Fig. 1A and Discussion section because the hydrophobicity of APD 2 was already published [4,13].
- Usually, the AMPs are heat-stable, and APD2 has a lot of similar features of AMPs, so you may test APD2 heat stability, if it is stable and still has antimicrobial activity, APD2 may be a relative of AMPs.
Author’s response
We are very grateful for your valuable comments. According to reviewer #2’s comment, we performed the thermal stability of AmAPD 2 using a PGN-binding assay (M&M section 2.7 and Fig. 3C).
- It would be better to add the BSA control group in Figure 4B binding assay?
Author’s response
We are very grateful for your valuable comments. We used E. coli sample without AmAPD 2 (E. coli-AmAPD 2) as a negative control.
- There are so many names in ref 10 and 12, can you just list the first 3 plus et al.?
Author’s response
We are very grateful for your valuable comments. According to reviewer #2’s comment, we revised it.
- Please add statistical significance information in figure2 legend.
Author’s response
We are very grateful for your valuable comments. According to reviewer #2’s comment, we added it.
- Do all the APD2 in Figure 1 have signal peptides? I just check AmAPD2, please check another two, and add the signal peptide information in figure 1A.
Author’s response
We are very grateful for your valuable comments. According to reviewer #2’s comment, we revised it.
- Line 33, delete “its expression is”.
Author’s response
We are very grateful for your valuable comments. According to reviewer #2’s comment, we revised it.
- In drone bee and queen, can pathogens upregulate APD2 expression?
Author’s response
We are so sorry about that and thank you very much for kind comment on that. Thus, further investigation will consider your valuable suggestion.
Once again, we greatly appreciate your valuable comments on improving the quality of the paper.